# Smart Glasses for Cultural Heritage: A Survey

Georgios Trichopoulos *, Markos Konstantakis and George Caridakis

Department of Cultural Technology and Communication, University of the Aegean, University Hill, 81100 Mytilene, Greece; mkonstadakis@aegean.gr (M.K.); gcari@aegean.gr (G.C.)
* Correspondence: gtricho@aegean.gr

**Abstract:** This paper presents a comprehensive survey on the utilization of smart glasses in the context of cultural heritage. It offers a systematic exploration of prevailing trends, the latest state-of-the-art technologies, and notable projects within this emerging field. Through a meticulous examination of diverse works, this study endeavors to categorize and establish a taxonomy, thereby facilitating a structured analysis of the current landscape. By distilling key insights from this categorization, the paper aims to draw meaningful conclusions and provide valuable insights into the potential future trajectory of SGs technology in the realm of CH preservation and appreciation.

**Keywords:** smart glasses; cultural heritage; museum; survey; technologies; case study; augmented reality; virtual reality; user experience

## 1. Introduction

Augmented reality (AR) technology is certainly not new and has found countless applications in the field of cultural heritage (CH) [1–5]. Smart Glasses (SGs) technology is also not new; thus, there are countless research projects on the use of SG in many fields of application, such as architecture, mechanical engineering, medicine, construction, electronic games, visual arts, etc. This research endeavors to compile a comprehensive collection of projects integrating AR SGs within cultural contexts. The search encompassed both implemented and tested projects and those existing solely in theoretical frameworks. These projects were categorized based on their targets and thoroughly analyzed. The aim is to equip readers with insights to initiate their own exploration in this field and potentially derive conclusions about emerging trends in technology use for the upcoming years.

But what are smart glasses? What categories of smart glasses do we target? The first construction of a head-mounted display dates back to 1968, under the name "The Sword of Damocles" [6–9] but the name Augmented Reality appeared in the 90's by Thomas Caudell [10]. Without going into more details about the history of AR SGs, in this survey we are referring to autonomous computing devices that can display augmented content on top of the actual image. They have their own processor and memory, storage medium, power source, camera, and a range of sensors such as a gyroscope and an accelerometer. They also have transceivers for communication such as Bluetooth, WiFi, and GPS. They can be wired or wirelessly connected to a personal computer and execute developer-generated code based on some API. The complexity of the construction of such a device, which will resemble as much as possible the glasses we wear in our everyday lives, is obviously great. We also want the glasses to have energy autonomy. To be able to be charged with electricity and then be able, without connection to any source, to operate independently for as long as possible. Finally, they should not be confused with virtual reality glasses that completely cut off images from the real world. It is necessary to maintain contact with the real space. The present research aims to find and highlight projects that utilize this technology in cultural spaces or in applications related to culture and cultural heritage.

The rest of this paper is structured as follows: Section 2, related work, forms the core of this study, outlining and analyzing various research works. Section 3 encompasses the

analysis of these works and a discussion on the subject. Section 4 presents a case study involving school students and finally, in Section 5 is the Conclusions.

## 2. Related Work

In the realm of SGs, a diverse array of projects has unfolded, showcasing the multifaceted applications of this transformative technology. From AR navigation systems to immersive educational experiences, these projects collectively exemplify the evolving landscape of SGs and their potential to reshape industries and daily life. In this survey, 37 projects or research papers focusing on the use of SGs in places of cultural interest were identified and analyzed. These projects were categorized into 7 groups based on their respective objectives.

### 2.1. Enhancing User Experience

Most papers explore the enhancement of user experience (UX) through the utilization of AR glasses in cultural spaces, such as museums and galleries. Notably, among the 38 projects under study, 11 aim to investigate the impact of specific technologies on visitors' experiences in these spaces. In our previous study [3], SGs were employed to enhance the UX by providing a visual representation of artworks and enabling interaction with them. These glasses support user engagement and incorporate elements of gamification. In a separate project by Yoon et al. [11], the authors conducted tests with real users to evaluate the use of augmented content through Microsoft HoloLens glasses at a specific Korean CH site. Through their application, they represent the object of interest in a three-dimensional (3D) form and display additional historical and informational elements around it.

In another approach by Brancati et al. [12], authors use the now-obsolete Google Glasses to enrich the experience of tourists. The head-mounted display is used to visualize all the most important cultural and tourist attractions directly highlighted in the user's field of view by means of overlaid points of interest (POI) icons. The user navigates through the information by means of tap and swipe touch gestures via a touchpad embedded in the side piece of the glasses. Vocal control is used to launch actions, such as "take a picture". Tilting the head up or down is used to enable or disable the display and to scroll through a list. The authors describe the experiments they conducted to assess the performance of their fingertip-based interface. They measured the performance in terms of movement time (MT), throughput (TP), and error rate (ER) by repeating measures on all twelve combinations of the three within-subject factors. They employed a four-factor mixed design in which the presence of the virtual pointer on the user's index fingertip is the between-subject factor, whereas the usage scenario (outdoors with sunlight, indoors with good lighting, and indoors with low lighting), target distance, and target width are the within-subject factors. The study involved twenty-four unpaid volunteers (14 males and 10 females), and the participants were randomly assigned to two different groups.

Through two projects [13,14], Rauschnabel et al. were trying to find out the extent of the acceptance of SGs technology by users as well as their concerns with privacy aspects. In the first project, the author proposes a conceptual model for understanding the psychological mechanisms that explain consumers' reactions to AR SGs. The model includes factors such as perceived usefulness, perceived ease of use, perceived enjoyment, social influence, and trust. The author suggested that these factors can help explain how and when consumers react to AR SGs. Additionally, other studies have investigated technology acceptance drivers for AR SGs, including personality traits and perceived health risks. Overall, the psychological mechanisms behind SGs use are complex and multifaceted, and further research is needed to fully understand them. While this study does not focus on CH specifically, it does demonstrate the potential for AR technology to enhance visitors' experiences in cultural institutions. In the second project, the authors present a study on the antecedents to the adoption of AR SGs. The study investigates the effects of utilitarian, hedonic, and symbolic benefits, as well as privacy risks, on users' intentions to adopt AR SGs. The results show that utilitarian and hedonic benefits positively influence adoption

intention, while privacy risks negatively influence it. Symbolic benefits have no significant effect. The study provides theoretical and managerial implications for the design and marketing of AR SGs and suggests avenues for future research. The file also includes details on the study's methodology, sample, and data analysis.

The CHATS project [15] utilized SGs to display digital information about artifacts of interest, including 3D models, images, and text that enhance users' knowledge and perception. The SGs also allowed users to interact with the virtual information displayed through AR techniques, such as manipulating the artifact to access different types of information based on the angle of view. Additionally, the SGs are used to digitally visualize data from the narratives, allowing users to watch characters "come to life" and narrate their stories through AR. On a similar path, according to the article by tom Dieck et al. [16], wearable SGs can enhance the museum experience for visitors by providing high-quality content, instant and personalized information, as well as links to other paintings. Additionally, the implementation of wearable AR SGs can provide visitors with a valuable, educational, and enjoyable experience. The use of AR allows for engaging content to be overlaid onto objects or artworks, creating a unique and hands-free opportunity for visitors to receive content while traveling or visiting public spaces. Museums that manage to create seamless AR applications, merging digital information with museum and art gallery (GLAMs) exhibits, are expected to be more competitive in the long term.

Another article that explored the potential of SGs in enhancing visitor experiences at museums was the one by Mason M. [17]. The author conducted a field experiment at the Robotics Gallery at the MIT Museum, where visitors were given Google Glass devices loaded with the Glassware prototype. The study found that visitors appreciated the ability to view exhibits while receiving information through the device, but there were some challenges with the mismatch of information and the length of written and video content. The authors identified six themes that emerged from the data, including looking at the object on display, digital content for SG applications, constant availability of information, direct access to content, navigation throughout the gallery, and sharing subjective experiences. Overall, the study suggests that SGs have the potential to enhance museum experiences, but more research is needed to optimize their use.

Bekele M.K. [18] proposed an interactive immersive map that allowed interaction with 3D models and various multimedia contents in museums and at historical sites. The proposed map can be applied to a specific Virtual Heritage (VH) setting in a predefined cultural and historical context and includes an interface that allows interaction on a map in a mixed reality (MR) environment. The researchers combined immersive reality technology, interaction methods, development platforms, mapping, and cloud storage services to implement the interaction method. Users could interact with virtual objects through a map that was virtually projected onto the floor and viewed through a HoloLens. The projected maps were room-scale and walkable, with potential global scalability. In addition to motion-based interaction, users could interact with virtual objects, multimedia content, and 3D models using the HoloLens standard gesture, gaze, and voice interaction methods.

In their study, Sargsyan et al. [19], the authors examined the growing use of AR (AR) as a prominent medium, particularly within museums, with the goal of enhancing the visitor experience. Understanding and analyzing visitor experiences within museum settings stands as a pivotal aspect for museums striving to offer enhanced services to society. Leveraging head-mounted cameras and eye-tracking headsets, this research explored the potential to analyze visitor experiences, especially with the integration of AR headsets within museum exhibitions.The focus revolves around an experiment conducted at the Jule Collins Museum of Fine Arts in Auburn, Alabama. The research measured visitor attention to exhibits using AR headset cameras, and it also collected exit surveys that reflected on the overall visitor experience. The paper provided a comprehensive depiction of the successes and limitations encountered when analyzing visitor experiences using AR headset cameras across various exhibits and scenarios. Furthermore, the study sheds light on pertinent issues, such as privacy concerns associated with employing AR headsets

in museum settings, and explores the impact of AR headsets on visitor experiences and behaviors. Through this analysis, the paper contributed insights into the nuanced dynamics of employing AR technology in museums and its implications for visitor engagement and privacy.

Finally, the article by Lee H. et al. [20] explored the benefits of utilizing smart devices and mobile applications to enhance the museum and exhibition experience. The authors propose a metaverse-based smart docent system that includes a digital docent, an avatar-participated mediawall, and other technologies to provide visitors with a personalized and interactive experience. The system includes features such as building a 3D map of the exhibition venue, providing storytelling inspired by the venue, and allowing visitors to create their own episode with respect to the article's contents. The authors also discuss the use of SGs, smart wristbands, and smart watches to provide VR and AR experiences. Overall, the paper proposes a direction where the museum and the exhibition can become a cultural space where people want to come again. SGs are mentioned in this document as a device that can provide VR and AR experiences for vivid docent services. The authors suggest that SGs can be used for mobile tour guidance and location-based AR games.

The concepts mentioned by projects in Section 2.1 are visualized into a sunburst diagram in Figure 1.

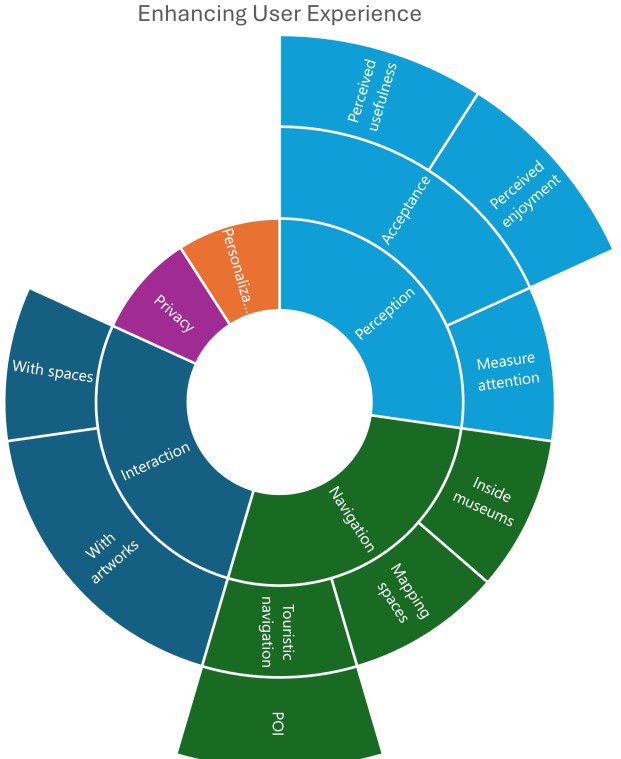

**Figure 1.** Sunburst diagram depicting the concepts about UX.

## 2.2. Comparing Mobile Devices vs. SGs

Some of the projects focus on comparing the UX between mobile devices (phones or tablets) and SGs. For example, Litvak et al. [21], discussed the strengths and weaknesses of an AR SG-based mobile guide compared to a more conventional smartphone-based guide. According to the results of the study, visitors were positive about using AR SG as a tool for exploring CH sites and learning about exhibits. The AR SG-based guide provided visitors with context-aware information regarding the POIs in their Field of View (FOV), which made it easier for them to navigate and learn about the exhibits. However, the study also highlighted several challenges that must be overcome before an AR SG-based system can be fully implemented for outdoor CH sites. These challenges include:

- *Device characteristics:* Participants experienced some challenges focusing on the near display while shifting focus between the displayed text and the far object ahead. The SGs are equipped with a replaceable dark tint visor that protects visitors from the direct sun and enables a better AR experience. Still, at times, participants had to look for a shaded spot ahead of them to improve the video display quality.
- *Usability problems*: Eleven participants felt that keeping their heads down to watch the smartphone "disconnects" them from the objects ahead and prevents them from enjoying their surroundings. Moreover, five participants stated that it was "too heavy" to hold the smartphone up high while interacting with the guide.
- *Technology acceptance factors*: Some participants found the colors of the smartphone's display more vivid and pleasant to the eye. Despite being larger than regular sunglasses, only two participants found the SGs to be bulky.

According to the survey article by Lee et al. [22], SGs use a variety of interaction methods that differ from those used in other wearable devices. For example, finger-worn devices and thumb-to-finger interactions are used as off-hand controllers for click-and-swipe gestures or other simple interactions. In contrast, SGs can use external devices such as smartphones, tablets, and laptops as input devices, which allows for more precise and responsive interactions. Additionally, SGs can use voice commands, head gestures, and eye tracking as input methods, which are not commonly used in other wearable devices. In any case, both handheld (smartphones) and wearable optical see-through devices have advantages and disadvantages, according to Serubugo et al. [23]. Handhelds are very accessible since most people own a smartphone, while SGs are hands-free. However, SGs offer a different approach to overcoming the limitations of traditional guides and allow for new exciting ways of presenting CH narrative content and making it interactive.

The concepts mentioned in the projects of Section 2.2 are visualized by a sunburst diagram in Figure 2.

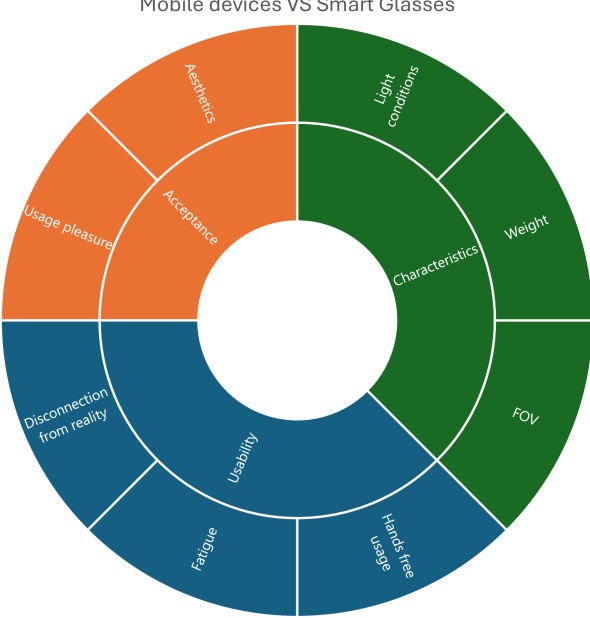

**Figure 2.** Sunburst diagram summarizing the concepts related to the comparison between mobile devices and SGs.

## 2.3. Ethics, Challenges and Privacy Risks

In previous projects, references have been made to issues of privacy violations arising from the use of SGs. However, there are more specialized studies specifically addressing this matter. Koelle et al. [24] worry about the privacy issues SGs create. This study explores the use of gestural interaction to enable bystanders to signal their preference for camera devices. The authors conducted a large-scale online survey to elicit potential opt-in and

opt-out gestures and investigate factors such as representativeness, social acceptability, and comfort level. They found that it is feasible to find gestures that are suitable, understandable, and socially acceptable. The study also addresses the ethical and legal implications of using such gestures for privacy mediation. Overall, the results suggest that gestural interaction can be a promising approach to enhancing privacy in public spaces.

Another article that discusses the ethical issues surrounding SGs is the one by Hofmann B. et al. [25]. The study provides an overview of the various ethical concerns related to the assessment, implementation, and use of this emerging technology. The authors discuss the potential impact of SGs on individual human identity and behavior, as well as the broader social and cultural implications of their use. The study also provides potential solutions or guidelines for addressing the ethical concerns related to SGs. The authors note that some references in the literature do not add value, as some references only mention that there are ethical issues without providing any analysis, and controversial statements may generate a lot of responses without adding value. In the work by Iqbal M.Z. et al. [26], the authors discuss the launch of SGs by Facebook (now known as Meta) in partnership with the Ray-Ban sunglasses brand's parent company, EssilorLuxottica. The SGs, called Ray-Ban Stories, have several technical features for entertainment and socializing but also come with ethical and privacy concerns. The article explores these concerns and their potential impact on public social interaction and daily life. The adoption of innovative wearable technologies is also discussed as a new trend.

The challenges arising from the widespread adoption of SGs technology are the subject of the paper by Kumar N.M. et al. [27]. These challenges include limitations in computing capabilities such as facial recognition, pattern recognition, and image processing, as well as insufficient power sources for energy-intensive tasks. Other challenges include a lack of strong network security and robust wired or Wi-Fi connectivity bandwidth, possible interruptions during communication, a lack of AR content, SG safety issues, and complex manufacturing. Additionally, privacy issues, a lack of strong regulations, a lack of use cases, and a lack of strong public acceptance and awareness of SG usage and development are also challenges. The concepts about ethics and privacy mentioned in Section 2.3 are visualized by a sunburst diagram in Figure 3.

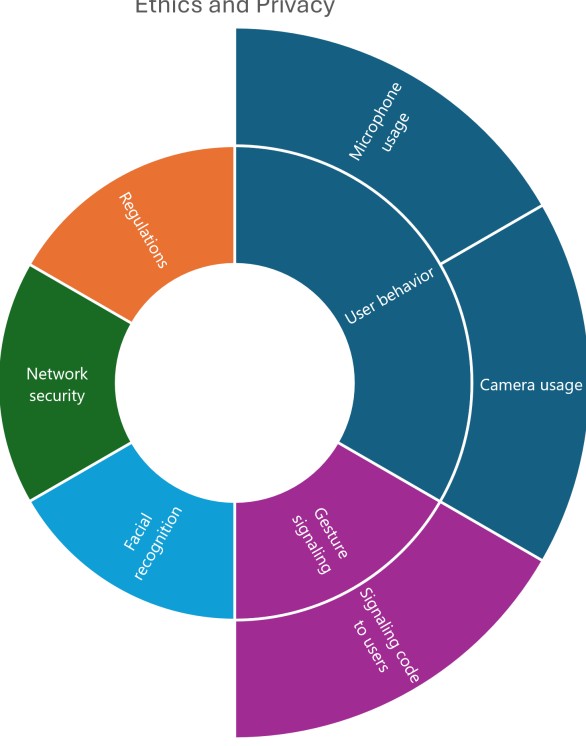

**Figure 3.** The diagram summarizes the concepts covered about ethics and privacy.

*2.4. Education and Storytelling*

In the article by Leue M.C. et al. [28] the authors explore the use of wearable computing and AR applications to enhance visitors' learning outcomes in art galleries. The study focuses on the effectiveness of the Google Glass Museum Zoom application in providing visitors with additional information on paintings, artists, and similar works. The results show that the majority of participants were able to improve their knowledge and understanding of the art and found the audio feature particularly helpful. The study also discusses the potential future applications of wearable computing and AR in art galleries and museums to further enhance visitors' learning experiences. In another article by Alkhafaji A. et al. [29] the subject of discussion is the design challenges for mobile and wearable systems to support learning at CH sites. It presents the results of a user study conducted to evaluate the SmartC mobile application prototype, which aimed to enhance visitors' engagement and learning experience at CH sites through the use of AR and smart eyeglasses. The study identified several challenges and issues related to interaction design, wearable computing, and the surrounding environment. The paper provides insights and recommendations for future research and development of mobile and wearable systems for CH sites.

According to Chen H.R. et al. [30], AR technology's application in museums primarily resides within scientific realms, with limited exploration concerning its potential in educational contexts. While some studies have touched upon the use of wearable devices for English language learning and their relation to students' learning styles and motivation, considerable gaps persist in this area. This research aims to address these gaps by examining the impact of integrating SGs into the English language-focused dinosaur exhibition at the Museum of Natural Science, specifically exploring their effects on learning outcomes and motivation. The study involved an experimental group utilizing SGs for learning strategies and a control group employing tablet-based learning strategies. Results revealed significant findings: (1) Learning effectiveness and motivation were notably higher among participants using SGs for the English language situated exploration game compared to those using tablet-based strategies. (2) The interaction between learning strategies and learning styles significantly influenced learning effectiveness; kinesthetic learners in the SGs group showed higher effectiveness, whereas visual learners using tablets exhibited greater effectiveness than auditory learners. (3) However, the interaction between learning strategies and learning styles did not significantly affect learning motivation, with visual learners demonstrating more sensitivity to motivation compared to auditory and kinesthetic learners. This research highlights the enhanced learning outcomes and motivation observed with SGs integration in educational museum contexts and emphasizes the nuanced impact of learning strategies on different learning styles.

The purpose of the wearable Mobile Reality (MR)-based mobile learning system created by Chin K.Y. et al. [31], is to integrate location-aware and virtual information technologies into physical environments, improve the information delivery method, and provide personalized learning support for each student. The system is designed to be a useful learning tool suitable for museum education. The purpose of the wearable MR-based mobile learning system is to integrate location-aware and virtual information technologies into physical environments, improve the information delivery method, and provide personalized learning support for each student. The system is designed to be a useful learning tool suitable for museum education.

Focusing on affective storytelling in cultural spaces, Dima M. [32] proposes a design framework that connects affective experiences with learning through three main pillars: interpretation, affective storytelling, and technology considerations. Interpretation involves providing curatorial guidance for visitors, outlining learning objectives, and guiding the design of stories, including chronology and the selection of archival material. Affective Storytelling pertains to the design of stories within the AR experience, emphasizing opportunities for affective, embodied meaning-making of history by visitors. Technology considerations consider the affordances of the device, exploring ways to support and

enhance storytelling while addressing potential challenges. In this framework, AR SGs serve as a new interaction method—hands-free, see-through, and capable of acquiring and utilizing sensorial information, including eye and gaze tracking. The use of AR SGs, with their see-through display, offers a more immersive experience compared to mobile devices, allowing visitors to maintain better awareness of their context when receiving information. The technology considerations pillar also considers how SGs' affordances can enhance storytelling. According to the source, an affective experience involves sensitive charges or felt intensities carried by words, sensations, thoughts, and emotions circulating in social spaces. In CH sites, affective experiences are crafted through the combination of stories presented amid the built environment, complete with its objects, smells, and sounds, and the hybrid space created to support storytelling. This setting allows for a felt, embodied, and enactive experience of meaning-making. The affective factor in constructing the story around a place contributes to experiential learning, a crucial element for heritage sites.

Testón A.M. et al. [33] employ digital avatars in the museum context to create natural and humanized museum visits using extended reality. The avatars function as storytellers, facilitating intuitive interaction with visitors and fostering a more immersive, intuitive, and seamless connection between the virtual and real worlds. The implementation involved a holographic guide using Hololens glasses at the Almoina Museum in Valencia. A functional prototype of an interactive human avatar served as a virtual holographic presence in the museum. The tour commenced with a mandatory five-minute introductory presentation, where a holographic avatar named Cleia introduced and explained the experience, along with instructions on interacting with the holograms. After learning about the system's operation, visitors could access four interactive content presentations, each lasting between five and ten minutes, positioned at four significant historical spots in the museum. Usability tests were conducted to assess the impact of the natural experience, yielding positive results. Users demonstrated a clear understanding of the tutorial, explaining how to activate the contents by simply pointing their heads for two seconds toward the interactive signals. No assistance was needed, and users moved naturally around the room. Users praised the application's attractiveness, citing the novelty of the media. They also noted its utility in comprehending the origins of the ruins and the associated construction. However, participants expressed a common concern about the device's limited field of view (FoV) at 34 degrees diagonal. Some visitors reported discomfort due to the weight of the glasses.

The concepts mentioned in Section 2.4 are visualized by a sunburst diagram in Figure 4.

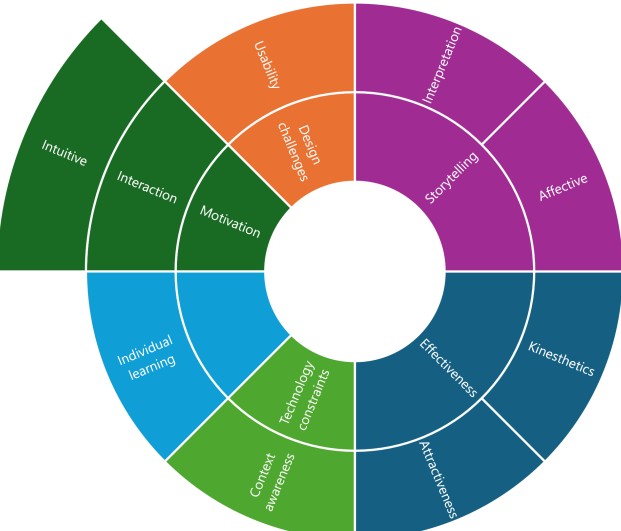

**Figure 4.** The diagram summarizes the concepts covered about education and storytelling.

## 2.5. Tourism and Cultural Routes

Obeidy W. et al. have contributed three studies that center on augmenting tourist experiences through the utilization of AR SGs. The TouristicAR application [34] provides context-aware information about UNESCO World Heritage sites in Malaysia. This information includes abstract information about the POIs and their location in relation to the user's current location. When a user selects a POI, a detailed view is shown with context-aware information about the selected POI. The context-aware information can include historical and cultural information, as well as other relevant information about the POI. In their other research [35], the proposed model focuses on measuring the effects of using SGs and AR on tourist experiences and behaviors in the context of CH tourism. The authors suggest that using AR and emerging technologies creatively as an educational, promotional, or marketing tool can form a positive image about the destinations and increase tourists' willingness to visit these destinations. These works were based on older research by the same authors [36]. The acceptance model presented in this text is designed to understand the usage behavior and visiting intention of tourists who use SG-based AR at UNESCO CH destinations in Malaysia. The model identifies external variables such as information quality, technology readiness, visual appeal, and facilitating conditions as key factors influencing visitors' beliefs, attitudes, and usage intentions. The model proposes that perceived ease of use and perceived usefulness have a positive effect on attitude, which in turn affects the intention to use and, consequently, the actual usage behavior and intention to visit behavior.

Han D.I.D. et al. [37] have authored an article that delves into the application of AR SGs in cultural tourism. The text provides a comprehensive examination of the use of AR SGs (AR SGs) within the cultural tourism industry. The authors present a detailed analysis of a case study conducted at a UK art gallery, investigating the integration of AR SG to enrich visitor experiences. Their findings indicate that AR SG offers additional information about paintings and cultural artifacts, with visitors generally expressing positive perceptions of its use in cultural tourism. However, the study acknowledges research design and data collection limitations, prompting the need for further exploration of how tourists perceive AR SG in different contexts to draw more generalized conclusions. The authors conclude by underscoring the importance of adopting a holistic approach in studying the integration of emerging technology within the tourism industry. In a different project by Ueoka R. et al. [38] the authors present an analysis of an AR (AR) event linked to the 'Godzilla at the Museum: Creative Tracks of Daikaiju' exhibition at the Fukuoka City Museum of Art in 2016. The event, named 'Godzilla meets 'F' museum', featured an AR backyard tour conducted as an extension of the exhibition. Through a survey-based approach, the authors examine the event's characteristics and implementation by gathering feedback from participants. The paper delves into the significance of integrating AR technology into art museum exhibitions and explores the novelty of such associated events based on the obtained survey data.

AR glasses are used in the Brescia-Brixia project [39] to provide visitors with a mixture of real and virtual experiences. Visitors are guided along an itinerary through the area of Brescia–Italy by an audio narrative available in various languages. The glasses overlay 3D reconstructions and dynamic videos onto the actual surroundings, rendering the ancient buildings as they were in the past while avoiding any technical or direct interventions on ancient walls or decorations. Visitors can enjoy a mixture of real and virtual and interact with the ancient monuments by looking at particular details or shapes, which function as markers and activate AR contents. The concepts mentioned in Section 2.5 are visualized by a sunburst diagram in Figure 5.

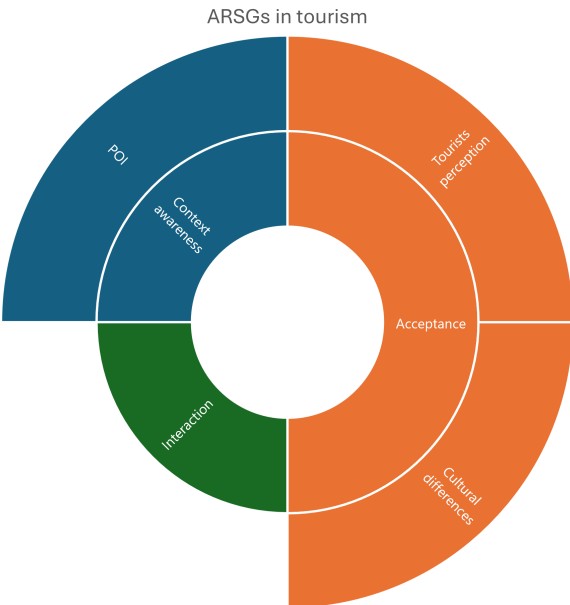

**Figure 5.** The diagram summarizes the concepts covered on tourism and cultural routes.

*2.6. Location Awareness and Navigation*

Three of the works under study on the use of SGs in cultural applications take advantage of the devices' cameras and sensors to locate visitors in closed spaces, as well as for navigation. In the paper by Kuflik T. et al. [40] the authors explore the potential of mobile AR technology in supporting visitors to CH sites, specifically in terms of navigation support. With the use of SGs, visitors can experience an engaging and intuitive way of interacting with their surroundings and learning about the history and significance of the CH site. The paper primarily focuses on the use of SGs for navigation support at CH sites. It discusses the potential benefits and challenges of using mobile AR technology to provide visitors with location-based information and real-time guidance without requiring them to carry and simultaneously view a hand-held guide. The paper also describes a prototype system that was developed to test the feasibility of this approach and outlines a planned evaluation to assess the usability of the system.

A novel localization system designed for museum settings utilizing SGs technology is introduced by Vu H.T. et al. [41]. Leveraging the pre-existing QR (Quick Response) codes embedded in artwork displays within the museum, the study proposes a hybrid approach employing an Inertial Measurement Unit (IMU), camera, and Wi-Fi radio signals for precise visitor localization and tracking. While the IMU offers direct self-motion data for target tracking, potential error propagation from gyro and acceleration biases necessitates additional refinement. To address this, the paper integrates a vision system, employing SGs to identify QR codes and utilizing camera geometry to establish the relative target location in relation to the QR code position. The study employs a Kalman Filter (KF) framework to refine visitor localization, combining IMU measurements and vision-based positioning. Subsequently, a Wi-Fi positioning system is employed within the defined region of interest (ROI) around the refined location. Utilizing a radio signal strength indicator (RSSI) and a trained radio map, a K-nearest neighbor (KNN) algorithm infers the visitor's precise location. Experimental results demonstrate the superior efficiency of this method compared to approaches relying solely on IMU, Wi-Fi, or a fusion of IMU and Wi-Fi methods.

NaviGlass [42] is an indoor localization system using SGs that uses inertial sensors predominantly and camera images for correcting position estimates. The proposed feature reduction method significantly reduces the computation time for image matching without compromising accuracy. NaviGlass was compared against Travi-Navi, a state-of-the-art localization system that uses both inertial and image sensors, and the results showed that NaviGlass has better image matching accuracy and achieved a mean localization error of

3.3 m, which is 64% less than that of Travi-Navi. The article also provides a review of related work and technical background, as well as a detailed description of the proposed method and its performance evaluation.The concepts mentioned in Section 2.6 are visualized by a sunburst diagram in Figure 6.

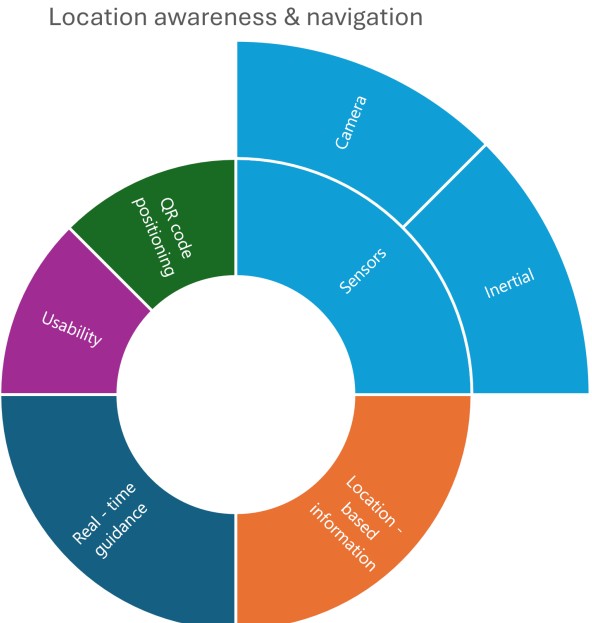

**Figure 6.** The diagram summarizes the concepts covered on location awareness and navigation using ARSG.

### 2.7. Surveys and Reviews

The final section of this survey encompasses prior surveys and reviews concerning the utilization of SGs technology across diverse applications. These resources not only provide insights into direct or indirect references related to cultural applications but also offer valuable conclusions. They serve as a foundation for generating new ideas and initiating projects within this domain. For example, Park S. et al. [43] provide a literature review of the use of Microsoft HoloLens technology in various fields, including engineering, medicine, education, and CH, and analyze the trends and functions provided by the technology. The paper authored by Vlachos A. et al. [44] delves into the examination of smart glass applications intended for AR. It outlines a comprehensive testing procedure encompassing installation, functionality, interruptions, and performance assessments. Notably, the paper critiques the limitations inherent in current AR applications, specifically those reliant on smartphones or tablets to display digital overlays. Emphasizing the examination of smart glass applications within the AR landscape, the paper offers valuable insights into the testing intricacies and the unique challenges associated with developing AR applications tailored for SGs.

The systematic literature review by Kim D. et al. [45] examines the current state of research on SGs in applied sciences. The review identifies and evaluates all available research relevant to SGs and categorizes them into seven application fields, including healthcare, computer science, social science, industry, service, culture, and tourism. The review finds that healthcare is the most active application field for SGs, followed by computer science and social science. The review also identifies the most commonly used products and operating systems for SGs and evaluates their effectiveness in various application fields. A discussion on the use of SGs in the culture and tourism fields is also included, and examples of research conducted on museum visitors are also provided.

Additionally, the review by Zuidhof N. et al. [46] provides a clear definition and characteristics of SGs, which have gained increased attention in both the research arena and the consumer market. The article sheds light on underexposed perspectives of SGs in

various fields and develops a definition that does justice to the state of the art of defining SGs. The article also discusses the challenges and future directions of SGs from the social sciences' perspective. The text includes a search methodology and thematic analysis to identify key themes and distinctive features of SGs.

## 3. Analysis and Discussion

It's evident that different writing teams or individual authors perceive SGs through their distinct backgrounds. Some adopt sociological perspectives, while others delve into intricate technical details regarding sensors and capabilities. There are also approaches emphasizing cultural implementation. But all researchers, without exception, agree that AR SGs can be an integral part of a museum or other cultural space and can enrich the experience of a visit. They remain a not-so-widespread and probably still expensive technology, but slowly and with steady steps, they are finding their way and their place in cultural applications.

The weaknesses and strengths of the technology become apparent from the above research projects. The glasses may remain from their first manufacture until today bulky, relatively heavy, and therefore tiring for long periods of use, strange in appearance, expensive, with limited energy operation time, and especially in the post-COVID-19 era, in limited stock. Since 2020, when the global crisis in electronics production began, due to the pandemic, the production of SGs has naturally stopped. Researchers, professionals, and ordinary home users have found it very difficult—if not impossible—to get hold of any model of glasses. There were few manufacturers in the field anyway; production stopped and demand increased as the US military placed the largest order of SGs at the time with Microsoft. The situation caused a large increase in the prices of products, and to date, in 2023, the situation has not completely normalized. In addition, SGs tend to show privacy risks. However, on the other hand, the glasses remain a very attractive technological implementation for the user-visitor. They allow freedom of hands in any activity and focus on the things that really matter. There is no need for museum visitors to hold their smart phone, which can become quite tiring when somebody needs to hold it up or focus on certain points for a long time, and there is no need to spend time using apps and manipulating them, which can be complex. Glasses offer a larger field of view than a phone for augmented content and offer a more natural interaction through the use of hands, iris, head movement, or voice commands. With some familiarity, they function as an extension of human functions, which means ease and freedom.

Table 1 depicts the various AR SG platforms used for the analyzed projects. In some research works, there is no clear mention of the AR SG implementation platform; in others, the authors have only stayed within the theoretical framework and did not proceed with any project implementation. Some of these SGs are now obsolete. Starting with Google Glasses, they were the first commercially available device but very quickly withdrew from the market and created a myth around them. The reason is that the device received a great deal of criticism, as there were concerns that its use could violate existing privacy laws, and a large consumer party was opposed to their use in public spaces. Google came back with two variations of the original model, but they are no longer supported. Microsoft glasses went from their first version to a second one, with better technical features and capabilities, and became a model for professionals such as engineers and architects. They work as a standalone system with a capable processor and a full array of sensors, but they remain the bulkiest and most expensive glasses on the market. The rest of the platforms in Table 1 only appeared in certain markets worldwide and were never broadly available. Newer models from the same manufacturers have already replaced some (if not all) of them. They apply the next-generation design, which wants SGs to look as much as possible like eyeglasses. This is a big step on the way to completely replacing smartphones with AR glasses.

**Table 1.** AR SG platforms used.

| No | AR SG Platform | Works | Remarks |
|---|---|---|---|
| 1 | Google Glass | Brancati N. et al. [12]<br>Mason M. [17]<br>Leue M.C. et al. [28]<br>Obeidy W.K. et al. [34] | Explorer model |
| 2 | Microsoft Hololens | Yoon Y.S. et al. [11]<br>Trichopoulos G. et al. [15]<br>Bekele M.K. et al. [18]<br>Chin K.Y. et al. [31]<br>Dima M. et al. [32]<br>Testón A.M. et al. [33]<br>Park S. et al. [43] | Hololens 1 & 2 models |
| 3 | Ray-Ban Stories | Iqbal M.Z. et al. [26] | |
| 4 | Epson Moverio | Vlachos A. et al. [44]<br>Serubugo S. et al. [23] | BT-350 & BT-200 models |
| 5 | Everysight | Litvak E. et al. [21]<br>Kuflik T. et al. [40] | Raptor model |
| 6 | Vuzix | Zhang Y. et al. [42] | M100 model |
| 7 | Sony Smart EyeGlass | Alkhafaji A. et al. [29] | |

Figure 7 shows the percentage of research works found in each category. As already mentioned above, the enhancement of UX is the first priority for researchers on the field. 11 out of 37 projects are related to this topic and constitute 30% of all researches. There are also three projects comparing UXs between mobile devices and SGs. While these projects constitute 8% individually, their thematic similarity allows us to incorporate this percentage into the previous category, further emphasizing the significance of UX enhancement. Education and tourism-culture are the second priorities of the researchers, with equal importance. Having both categories from 6 works in the total of 37 research papers, they each occupy a percentage of 16%. Surveys and reviews on the topic of AR SG share the same percentage of 11% with the research papers on ethics and privacy concerns related to the use of AR SG. Specializing in exploiting the potential of SGs to provide location awareness in cultural spaces for visitors, 3 projects with a rate of 8% complete the total projects.

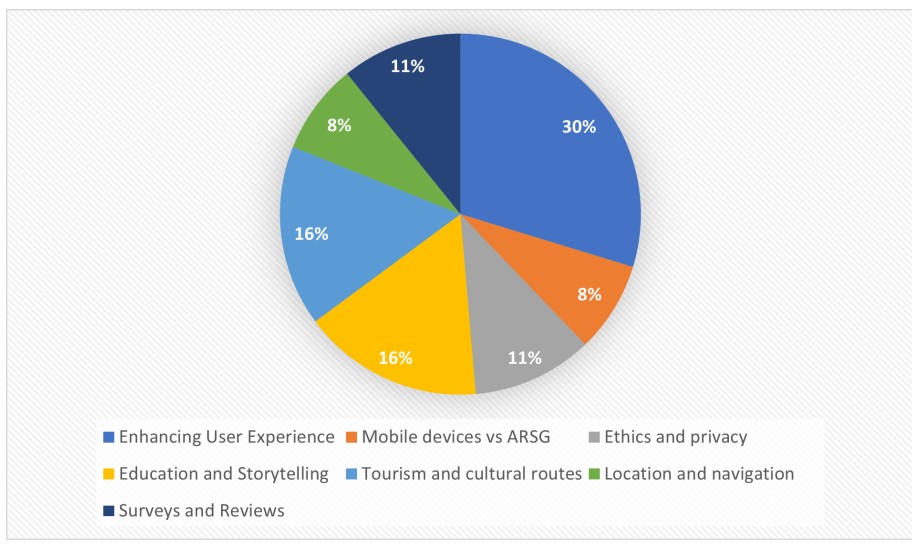

**Figure 7.** Percentage of works in each category.

In Table 2, projects—excluding Surveys and Reviews, which encompass a range of project types—are categorized based on their target content. The table columns were derived from text analyses, after identifying commonalities and distinctions among project contents. Three symbols are employed in the table to indicate the level of project involvement in specific categories. A solid black circle denotes the primary focus of the text, a half-black circle signifies moderate engagement, while a transparent circle indicates references to a particular category. The order of the works in the table is the same as the order in which they were mentioned in the text above, thus maintaining numbering and consistency, with the aim of making the text easier to read.

**Table 2.** Research categorization.

| No | Works | UX | Mobile Devices vs. SGs | Ethics & Privacy | Education & Storytelling | Tourism & Culture | Location Awareness |
|---|---|---|---|---|---|---|---|
| 1 | Trichopoulos G. et al. [3] | ● | | | ◐ | | |
| 2 | Yoon Y.S. et al. [11] | ● | | | | ◐ | |
| 3 | Brancati N. et al. [12] | ● | | | | ◐ | |
| 4 | Rauschnabel P.A. [13] | ● | | ○ | | | |
| 5 | Rauschnabel P.A. et al. [14] | ● | | ◐ | | | |
| 6 | Trichopoulos G. et al. [15] | ● | | | ◐ | | |
| 7 | tom Dieck M.C. et al. [16] | ● | | | ◐ | | |
| 8 | Mason M. [17] | ● | | | | | |
| 9 | Bekele M.K. [18] | ● | | | | | |
| 10 | Sargsyan N. et al. [19] | ● | | ○ | | ◐ | |
| 11 | Lee H. et al. [20] | ● | | | ○ | ◐ | |
| 12 | Litvak E. et al. [21] | ◐ | ● | | | | ○ |
| 13 | Lee L.H. et al. [22] | ◐ | ● | | | | |
| 14 | Serubugo S. et al. [23] | ◐ | ● | | | | |
| 15 | Koelle M. et al. [24] | ◐ | | ● | | | |
| 16 | Hofmann B. et al. [25] | | | ● | | | |
| 17 | Iqbal M.Z. et al. [26] | | | ● | | | |
| 18 | Kumar N.M. et al. [27] | | | ● | | | |

**Table 2.** *Cont.*

| No | Works | UX | Mobile Devices vs. SGs | Ethics & Privacy | Education & Storytelling | Tourism & Culture | Location Awareness |
|----|-------|----|------------------------|------------------|--------------------------|-------------------|--------------------|
| 19 | Leue M.C. et al. [28] | | | | ● | | |
| 20 | Alkhafaji A. et al. [29] | | | | ● | | |
| 21 | Chen H.R. et al. [30] | | | | ● | | |
| 22 | Chin K.Y. et al. [31] | | | | ● | | ◐ |
| 23 | Dima M. [32] | | | | ● | | |
| 24 | Testón A.M. et al. [33] | ◐ | | | ● | | |
| 25 | Obeidy W.K. et al. [34] | | | | | ● | |
| 26 | Obeidy W.K. et al. [35] | | | | | ● | |
| 27 | Obeidy W.K. et al. [36] | | | | | ● | |
| 28 | Han D.I.D. et al. [37] | | | | | ● | |
| 29 | Ueoka R. et al. [38] | ◐ | | | | ● | |
| 30 | Morandini F. et al. [39] | ◐ | | | | ● | |
| 31 | Kuflik T. et al. [40] | ○ | | | ○ | | ● |
| 32 | Vu H.T. et al. [41] | | | | | | ● |
| 33 | Zhang Y. et al. [42] | | | | | | ● |

● = Primary focus of the text,   ◐ = Moderate engagement,   ○ = Simple references.

Thus, and starting from the most populous category, we observe that the largest percentage of projects have as their main objective or make clear references to research around the experience of using SGs in cultural spaces. It is a topic that is of particular concern to researchers and based on their texts, glasses offer a more natural interaction with the space and can enrich the visiting experience, but there are always privacy issues that need to be resolved [13,14,19]. This experience has been more specifically studied in educational settings [3,15,16,20] and in tourism applications [11,12,19,20]. The experience of using smart glasses has been compared to that of using smart phones [21–23], as the former technology aspires to replace, even partially, the latter.

The issue of privacy violations related to the use of smart glasses is highly significant and has been present since the inception of these glasses in the market. The built-in camera, capable of capturing photos and videos while the user is in motion, without always signaling its use since the user's hands remain free, raises valid concerns. Ensuring compliance with ethical and legal standards is left to the user's discretion, without effective

monitoring. Thus, some researchers focus specifically on this topic, in a specialized way for cultural spaces [24–27].

AR technology in general contributes to a very large extent to the upgrading of educational processes and enriches the learning experience. This is something that has been studied for many years by a large number of researchers [47–51]. On this basis, specialized research on the use of SGs in museum education could not be lacking [28–31]. The impact of digital storytelling in education is also huge, in terms of student engagement and satisfaction, ease of learning and memory retention [52]. Thus, some research focuses on the application of digital storytelling methods for museum education, using augmented reality and smart glasses as tools for immersion and greater engagement [32,33].

An important share of projects deals with tourism and cultural routes. It is natural as people visiting cultural heritage sites is one of the foundations on which tourism around the world rests. The use of SGs in sightseeing routes or in the highlighting of monuments is something that is applied more and more, with the goals of impressing visitors, highlighting historical elements that may no longer exist, encouraging the interactive participation of tourists, creating richer experiences for them and boosting the revenue influx. In the present research, 6 research projects focusing on the use of SGs in tourism applications have been selected [34–39], but there are an additional 4 projects that are partially involved in this area [11,12,19,20].

A more specialized and technical issue is locating the visitor in a museum. This is useful in forming a profile for the visitor and providing personalized recommendations. As classic geolocation technologies cannot work in closed spaces—without communication with geospatial satellites—the issue has long been a concern of researchers. The SGs come to provide solutions, if the sensors they carry and the built-in camera are used properly. According to the above, in Table 2 three research projects have been placed that are specialized in position locating through SGs [40–42] and two more that refer more or less to the subject [21,31].

## 4. Case Study: Experimenting with School Students

In an exploration of emerging technologies, our study delved into the immersive realms of AR and VR by employing two cutting-edge AR glasses—the Microsoft HoloLens 2 and the Vuzix Blade—alongside the Oculus Rift S VR headset. The focus of our investigation was to introduce teenagers to these devices, allowing them to engage firsthand with the transformative capabilities of AR and VR. Through systematic tests and experiential interactions, we aimed to draw preliminary conclusions regarding the perceived satisfaction of the participants during the use of these technologies. By bridging the gap between curiosity and technological integration, our research contributes valuable insights into the evolving landscape of AR and VR, particularly in the context of teenagers' experiences with these immersive devices.

The tests were carried out in 3 phases, with different groups of teenagers. In the first phase, which lasted for four weeks in April and May 2022, students from the Music School of Mytilene on the island of Lesvos, Greece, were invited to test the AR devices Microsoft HoloLens 2 and Vuzix Blade, as well as the VR device Oculus Rift S. The devices were used by a total of 103 students, aged 12 to 18, of whom 55 were girls and 48 were boys. The second phase of the tests took place in October 2022, for two weeks, in the same school and with the same devices, with a group of 30 newly entering students, aged 12–13 years, of which 17 were girls and 13 were boys. The last phase of the test took place in a different school, in October 2023. 123 students of the Model General Lyceum of Mytilene, Greece, 67 boys and 56 girls, aged 15–17, tested the same devices over a period of 3 weeks. The above is summarized in the chart of Figure 8. There were 3 Vuzix devices, 2 Oculus devices and only one HoloLens available, 6 devices in total. All participants tested all devices for a time not exceeding the 7-min limit, and then had to pass the device to the next student. During the exchange of devices, the devices were cleaned and disinfected at the same time.

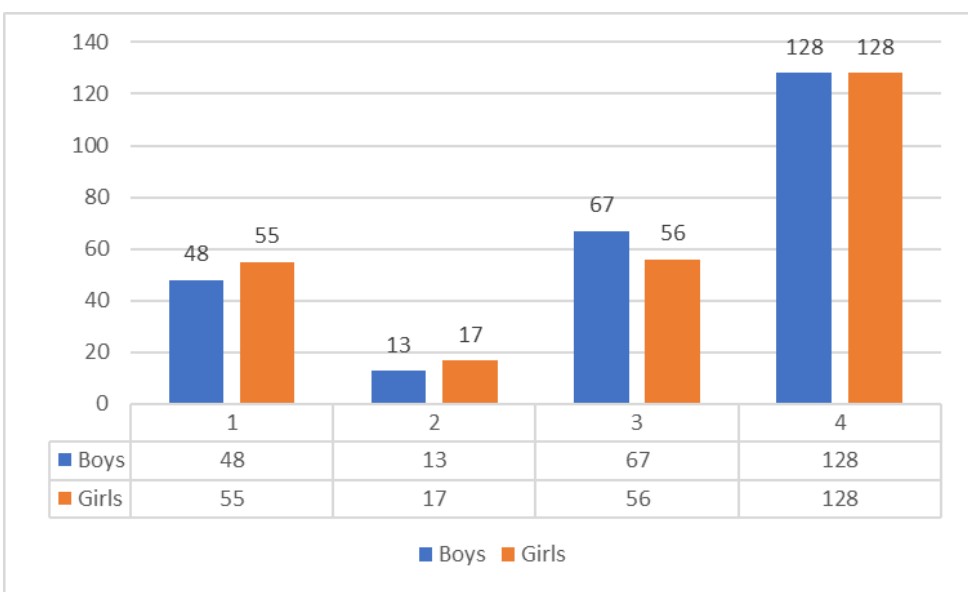

**Figure 8.** Number of students who participated in the tests.

The tests were carried out during the Informatics class. The time available each time for each group of students was the 45 min of the lesson, and there was always preparation time, time to solve technical problems, time to disinfect the devices, and the last minutes were always devoted to discussions. Students who weren't using the devices at any given time could watch the reactions of the students who were using them, since everyone was always together in the classroom. With the above devices students had to take specific steps that were common to all. In Microsoft HoloLens 2, they started with the Tips app which is pre-installed on all the devices [53], where they learned the basics of operating the device and then used a custom app titled Museum Guide. In this simple application, the 3D model of an animal skeleton (an otter or a mosasaur) is displayed in front of the user, and the user can grab it, move it, rotate it, zoom in or out, walk around of (or in) it and study details of the skeleton. At the same time, a short narration about the animal is heard and there is the possibility to increase or decrease the volume of the sound through a slider (Figure 9). These skeletons can be found inside the Museum of Paleontology and Geology of the National and Kapodistrian University of Athens and their 3D digital twins were created using photogrammetry and laser scanning, as part of another research project [54]. The Museum Guide app was built in Unity.

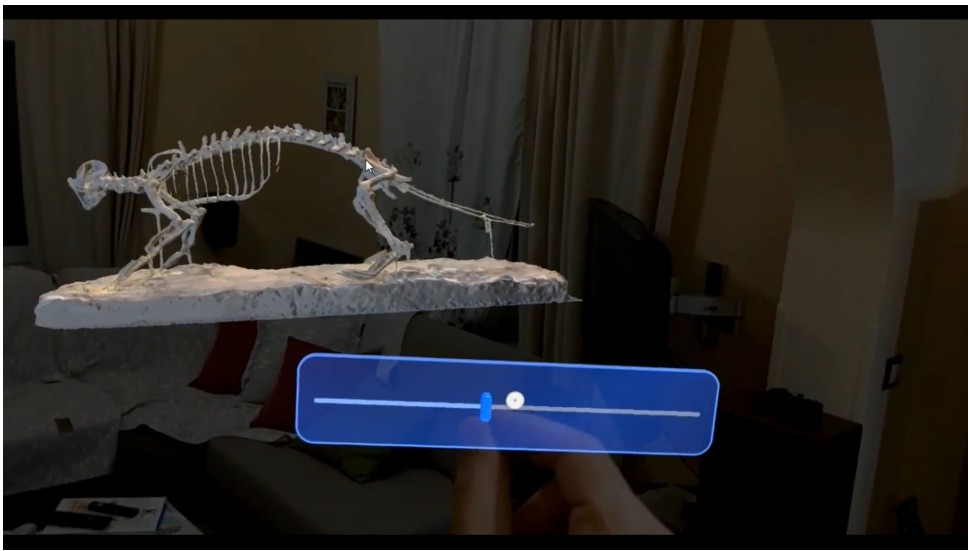

**Figure 9.** The Museum Guide app.

In the Vuzix Blade glasses, users had to navigate the device's menus to get used to using it and then open the Dino Hunt game, which can be downloaded for free at the Vuzix App Store [55]. The device supports manipulation through its right arm, with taps and swipes. The basic supported gestures are one-finger tap and one finger swipe forward and backward but there are more advanced gestures that work with two fingers, as illustrated in Figure 10. The image is taken from the official user manual of the device [56]. The device also has a gyroscope which allows it to respond to the movement of the user's head. This particular game was chosen because it is simple to use, requires no special instructions to get started, and takes advantage of the device's gyroscope and control capabilities, exposing the player to all of the device's interaction capabilities. Plus, it pushes the device to its processing limits and it's fun!

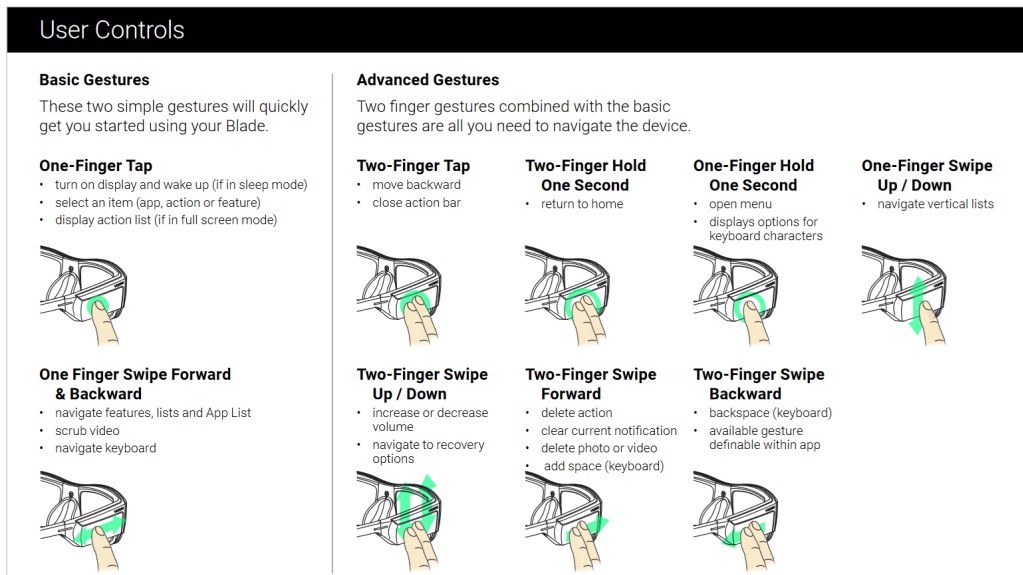

**Figure 10.** The gestures supported by Vuzix Blade SGs. Image taken from the user manual [56].

The Art Plunge application was used on the Oculus Rift S device [57]. This is an application in which accurate VR interpretations of 5 famous works of art have been implemented. The user can be inside the space of the painting and experience its every detail. User interaction with the app is limited to entering and exiting a painting and is a sit-down experience in a confined space. During the first phase of testing, with the first group of students, another application that required movement within a larger space was also tested. For the safety of the students, a space 5 m long and 3 m wide, free of obstacles, had been prepared and demarcated. The application was the game called Richie's Plank Experience [58], which is an intense experience for the user, since it can create intense feelings. The study of these feelings is not in the scope of this survey.

Starting with a comparison between the devices, the user experience is very different in each case. HoloLens is controlled by gestures in the air. The user can handle the augmented content as if it is real, and the focus of the gaze can be important. Voice instructions can also be combined along with the gestures for a more complete user engagement. In the Vuzix Blade, moving the head left and right is important in handling, but most actions require the user's right hand to be placed on the device. Part of the functionality of the device relies on its interconnection with a smart phone and the corresponding application.

The assessment of the UX had an informal form and was done in three ways: (a) Observation. Reactions of the teenagers were noted, when they used the devices themselves and when they observed their classmates, (b) Discussion. Dialogues with students during the tests and at the end of each lesson helped to draw conclusions, (c) Quiz. The Kahoot application (https://kahoot.com/ (accessed on 13 March 2024)) was used to create a quiz game with questions related to the use of devices and ways of interacting with them. The answers to the questions showed, firstly, that the students had understood the way

of operation and the capabilities of the devices, and secondly, that their experience was completely positive. In the comparison between the experience of VR and AR, they found VR more impressive but at the same time they did not like the idea of losing touch with the rest of the class. In these conditions where other students could interact with the user of the device, some students felt uncomfortable losing contact with the environment. In using AR glasses, participants enjoyed the ability to simultaneously observe the space around them and communicate with the rest of the class, and the freedom to walk around the space.

Comparing the AR devices to each other, the largest percentage of students (about 81%) found the experience of using HoloLens more interesting as they were impressed by the way it operated with gestures in the air. The rest chose Vuzix because "it's easier to wear on the road", "it connects to the mobile phone", "it's lighter and discreet", "it's very easy to use, just like operating a mobile phone". The above are reflected numerically in the table of Figure 11.

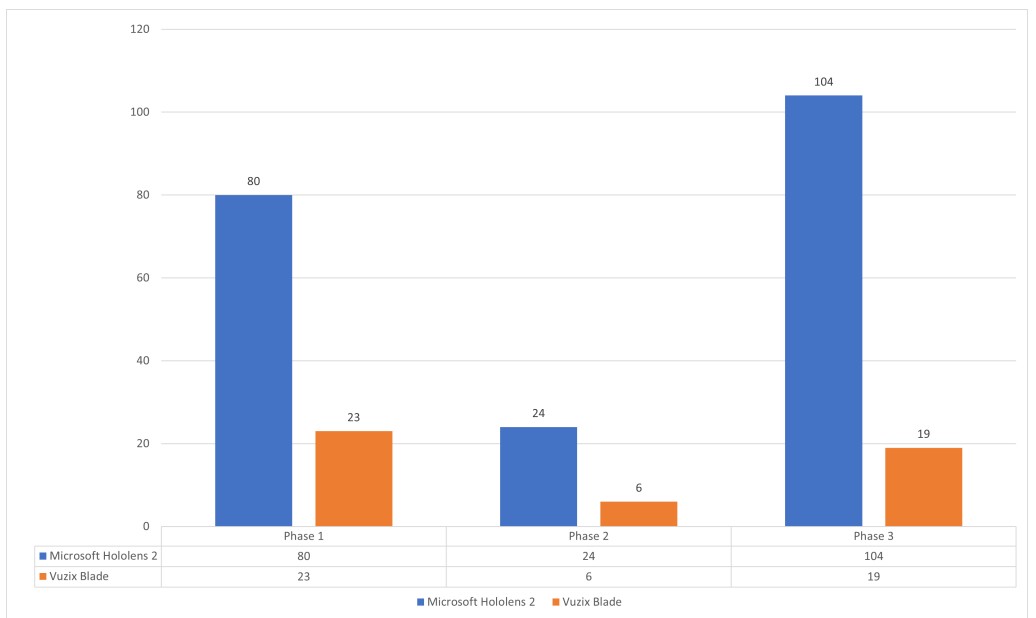

**Figure 11.** Preference between the devices amongst students.

There were a few cases (about 2% of all attendees) who refused to use any of the devices, using feedback such as "I've done it before", "I have a problem with my eyes" or "I don't like it". Teenagers who wore glasses due to myopia or other conditions faced substantial difficulties in using the AR devices. In Hololens the problem was not great as the headset, with a suitable adjustment, allowed the spectacles to be worn underneath. But in the case of Vuzix, the device was impossible to use, as users with a certain visual condition could not see the image clearly. In any case, the acceptance of the devices and technologies was great. A common question in the discussions was "how much do they cost?" and a common comment that followed was "they are very nice but also very expensive".

## 5. Conclusions

Augmented content and physical interaction create greater user engagement and enhance the experience of visiting a museum or other cultural site. The glasses also offer more capabilities such as tracking and orientation in closed spaces—where GPS cannot work. So why haven't we all abandoned our smart phones and been using SGs already? Let's try to give some answers, not necessarily in order of importance. Some of the answers come from literature research, and some others from the authors' experience, through experimentation and participation in previous research projects.

Though aesthetics might not hold significance at the research level, its role in a product's commercial value is paramount. Drawing a comparison to the smartphone market, Apple's introduction of the first smartphone in 2007 heavily emphasized the appealing

design of the product. This approach, combined with the new capabilities of interacting with the device, sparked a major technological and commercial revolution, quickly gaining mass acceptance worldwide and putting smartphones in people's hands on a global scale. The story with smart glasses is of course somewhat different. People have already been wearing glasses for centuries. They had already acquired a certain aesthetic of what beautiful glasses should look like. However, placing all these subsystems that smart glasses require, inside the thin frame that holds the lenses, is something that remains impossible to this day. There is great progress in this direction, but the final goal has not yet been technologically reached. An attempt is made, in a clever way, to fit a computing system into the infinitesimal space that can be offered by some cavity in the frame of the glasses. In addition, the battery has to fit in the same space and this is a separate challenge in itself.

So the issue of energy is also very important. Microsoft's Hololens offer quite a long autonomy but in exchange for their large volume. Conversely, glasses that tried to maintain smaller proportions have a much lower ability to stay active. In experiments with Vuzix Blades, their operation could not last for more than 20 min, which is probably not enough time for a visit to a museum. As an extension of energy consumption, the devices show an increase in their temperature, mainly in the spots where the processor and the battery are located. During the case study, some students felt uncomfortable by the extreme heat of the device on their head. It is a device design problem that is under study [59].

The issue of weight also exists. Smart glasses are clearly heavier than eyeglasses. Thus, prolonged use can cause symptoms of fatigue in the support points, such as for example in the upper part of the nose. Hololens strap on top of the head and tighten at the back, so they feel more like a hat than glasses. At this point, we will also add the reflection on the risk to the health of the users. These are devices that emit frequencies in different bandwidths in order to communicate with the environment. So we have the microwave frequencies of 2.4 GHz and 5 GHz for WiFi, while Bluetooth also works in the same 2.4 GHz band. GPS satellites broadcast on at least two carrier frequencies between 1575.42 MHz, 1227.6 MHz and 1176 MHz. In fact, if the manufacturers proceed to install SIM cards for the mobile networks, then we will also add the frequency bands of the 4G and 5G networks which may differ in each continent and region. These transceivers are in contact with the user's head during use of the device and the health burden rates of the users are being studied [60] and efforts are being made to redesign the devices with lower levels of radiation [61], but it is too early to draw any firm conclusions. What is certain is that the issue will have to be studied before SGs gain wider acceptance.

In addition, the SGs brought to the fore-again-the issue of the violation of privacy [14,24,62]. When mobile phones invaded our lives and our personal space this reflection started for the first time as our geographical location and the way and time in which we move every day became known. The users themselves broke down any form of privacy, self-posting photos that they could easily take from their phone and sharing their location constantly on social networks. The issue with SGs becomes more complex as it becomes easier to discreetly capture and share photos or videos of others without their consent. So if to all of the above, we add some safety issues that still exist with the use of these devices [62,63] but mainly, their cost which remains high and the availability which is low in relation to the demand, we can perhaps understand why not everyone has a device at home yet and why we don't see people walking down the street wearing SGs.

During our tests, students who have an eye condition such as myopia, astigmatism or farsightedness and wear glasses or contact lenses every day, had difficulty using the SGs we had available. The manufacturers of the devices require users to order their own prescription lenses to fit their eyes exactly, which is another reason why the same glasses may not work for everyone. Beyond all the above concerns and difficulties, SGs remain an attractive technology for use in cultural and non-cultural spaces. Despite their existence for decades, they are still in development and it may be some time before we see perfected models. Nevertheless, our experiments showed that they brought teenagers joy, often excitement. The experience of using them was very positive and users got used to handle

them without instructions, with gestures and voice commands (in Microsoft HoloLens) or operating from the arm of the glasses (in Vuzix Blade).

From the side of developers, it is found that developing apps for AR devices is not the easiest task. Documentation is often lacking, developer and forum communities are very small, so programming for these devices is still in an experimental stage. It is still a limited market and this can be seen from the very small amount of ready-made applications that a user-owner of such a device can download or buy from the official online stores. In addition, each manufacturer adopts a different programming language, a different programming environment, creates its own API, and there is no established design and implementation standard, which makes application development more complicated.

**Author Contributions:** Conceptualization, G.T.; methodology, G.T.; software, G.T.; validation, G.T. and M.K.; formal analysis, G.T.; investigation, G.T.; resources, G.T.; data curation, G.T.; writing—original draft preparation, G.T. and M.K.; writing—review and editing, G.T. and M.K.; visualization, G.T.; supervision, G.C.; project administration, G.T. and M.K.; funding acquisition, M.K. All authors have read and agreed to the published version of the manuscript.

**Funding:** This research received no external funding.

**Conflicts of Interest:** The authors declare no conflict of interest.

## Abbreviations

The following abbreviations are used in this manuscript:

| | |
|---|---|
| SGs | Smart Glasses |
| AR | Augmented Reality |
| AR SGs | Augmented Reality Smart Glasses |
| API | Application Programming Interface |
| CH | Cultural Heritage |
| VR | Virtual Reality |
| GLAMs | Galleries, Libraries, Archives and Museums |
| MR | Mixed Reality |
| UX | User eXperience |

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
