# Peer review of "Smart Glasses for Cultural Heritage: A Survey"

_heritage, doi:10.3390/heritage7030078_

Round 1

Reviewer 1 Report

Comments and Suggestions for Authors

Dear authors,

I commend you on the depth and effort invested in your survey on smart glasses for cultural heritage. The extensive 20 plus pages content reflects your dedication. However, for the sake of readability, please consider refining the paper's length. Streamlining and enhancing the editing in certain sections could significantly improve accessibility, ensuring your insight are more easily digested by readers.

Overall, great work, and a bit of fine-tuning could make it even more impactful.

I appreciate the thoroughness of your references, but I noticed a bit of self-promotion in some instances. Regarding the first reference "A Survey on Augmented Reality Application in Cultural Heritage", I wonder if you could shed light on any substantial differences between this paper and  "Smart Glasses for Cultural Heritage: A survey". Unfortunately I don’t have access to the mentioned paper at the moment, and considering the time constraints, even if I did, I might not have been able to delve into it thoroughly. Your clarification on this matter would be much appreciated.

Comments on the Quality of English Language

Dear Authors,

Your paper possesses valuable insights, and the language is proficient.

However, there is a slight challenge in digestibility. Consider a smoother flow throughout the entire paper for improved clarity and accessibility

Author Response

Firstly, we would like to thank Reviewer 1 for his/her effort in reviewing our manuscript. We really appreciate the careful review and constructive suggestions. In what follows, we try to address all the points raised in the review. It is our belief that the manuscript is now substantially improved after making the suggested edits.

Attached are the responses for each comment.

Reviewer 2 Report

Comments and Suggestions for Authors

There is a lack of conlusions. This issue needs to be adressed!

Author Response

Firstly, we would like to thank Reviewer 2 for his/her effort in reviewing our manuscript. We really appreciate the careful review and constructive suggestions. In what follows, we try to address all the points raised in the review. It is our belief that the manuscript is now substantially improved after making the suggested edits.

Attached are the responses for each comment.

Reviewer 3 Report

Comments and Suggestions for Authors

This paper presents a comprehensive and well-structured exploration of smart glasses (SGs) within the cultural heritage sector, offering an insightful analysis of current trends, state-of-the-art technologies, and notable projects in the field. By systematically categorizing existing works, the authors effectively draw meaningful conclusions about the potential and future trajectory of SG technology in enhancing the preservation and appreciation of cultural heritage. They adeptly highlight the promising applications of SGs in augmenting cultural heritage experiences, while also candidly addressing the practical, technical, and ethical challenges hindering broader adoption.

The taxonomy developed for this study is particularly instrumental, providing a clear and organized framework that enriches the analysis of SG applications in cultural heritage. This framework not only offers a comprehensive overview of the current landscape but also underscores significant opportunities for future research and development.

While the paper is commendable for its clarity, depth of analysis, and interdisciplinary perspective, future work could benefit from expanding the comparative analysis between different SG technologies and their specific applications in cultural heritage. Additionally, a deeper exploration into the cultural, social, and ethical implications of these technologies could further enrich the discourse, providing a more holistic view of their impact.

Overall, this paper makes a compelling case for the potential of smart glasses in cultural heritage, skillfully navigating the complex interplay between technological advancement and cultural preservation. It is a valuable addition to the literature, offering both a solid foundation for future research and practical insights for the deployment of SGs in cultural contexts.

Author Response

Firstly, we would like to thank Reviewer 3 for his/her effort in reviewing our manuscript. We really appreciate the careful review and constructive suggestions. In what follows, we try to address all the points raised in the review. It is our belief that the manuscript is now substantially improved after making the suggested edits.

Attached are the responses for each comment.

Reviewer 4 Report

Comments and Suggestions for Authors

Dear authors,

Congratulations on this very interesting and well-structured manuscript. The following two reflections are aimed at boosting the readability of your article:

-Seen the technical nature of the article, it would be useful to visualise the tools and their applications in the first part of the articles (P2-9)

-It is suggested to change the title of section 5 which is currently titled "Discussion" to conclusions and add the word discussion to section 3 so it could become analysis and discussion.

Author Response

Firstly, we would like to thank Reviewer 4 for his/her effort in reviewing our manuscript. We really appreciate the careful review and constructive suggestions. In what follows, we try to address all the points raised in the review. It is our belief that the manuscript is now substantially improved after making the suggested edits.

Attached are the responses for each comment.
